# Seasonal and periodic patterns of PM2.5 in Manhattan using the variable bandpass periodic block bootstrap

Yanan Sun*, Edward L. Valachovic

Department of Epidemiology and Biostatistics, College of Integrated Health Science, University at Albany - State University of New York, Rensselaer, New York, United States of America

* ysun7@albany.edu

## Abstract

Air quality is a critical component of environmental health. Monitoring and analysis of particulate matter with a diameter of 2.5 micrometers or smaller ($PM_{2.5}$) plays a pivotal role in understanding air quality changes. This study focuses on the application of a new bandpass bootstrap approach, termed the Variable Bandpass Periodic Block Bootstrap (VBPBB), for analyzing time series data which provides modeled predictions of daily mean $PM_{2.5}$ concentrations over 16 years in Manhattan, New York, the United States. The VBPBB can be used to explore periodically correlated (PC) principal components for this daily mean $PM_{2.5}$ dataset. This method uses bandpass filters to isolate distinct PC components from datasets, removing unwanted interference including noise, and bootstraps the PC components. This preserves the PC structure and permits a better understanding of the periodic characteristics of time series data. The results of the VBPBB are compared against outcomes from alternative block bootstrapping techniques. The findings of this research indicate potential trends of elevated $PM_{2.5}$ levels, providing evidence of significant semi-annual, tri-annual, and weekly patterns missed by other methods.

## Introduction

Environmental factors have a significant impact on public health [1–6]. Understanding environmental elements, such as air quality, is crucial for studying and preventing some health issues and promoting health at a population level. A key indicator for monitoring air quality is the particulate matter with a diameter of 2.5 micrometers or smaller ($PM_{2.5}$). Long-term $PM_{2.5}$ monitoring is useful for assessing the health risks associated with air pollution.

Long-term exposure to PM2.5 has been linked to various health problems, including nonaccidental mortality and several types of cancer [1–6]. The conclusions of the International Agency for Research on Cancer (IARC) indicate that exposure to

**Data availability statement:** DOI: https://doi.org/10.6084/m9.figshare.29132738

**Funding:** The author(s) received no specific funding for this work.

**Competing interests:** The authors have declared that no competing interests exist.

ambient air pollution and particulate matter is associated with an increased risk of lung cancer [2]. One study also investigated lung cancer in the older adult population in the US, indicating that long-term exposure to $PM_{2.5}$ is significantly associated with increased lung cancer mortality [3]. This finding aligns with the conclusions of the IARC. In addition to lung cancer, researchers have found there was a correlation between $PM_{2.5}$ exposure and an increased occurrence of estrogen receptor-positive (ER+) breast cancer, in contrast to estrogen receptor-negative (ER-) breast cancer, suggesting a potential endocrine-related mechanism [4]. Beyond cancer, cardiovascular disease (CVD) also has a strong association with long-term exposure to $PM_{2.5}$, representing another significant health risk linked to this particulate matter [5–6]. Given the numerous diseases linked to $PM_{2.5}$, studying the periodic characteristics of this is crucial. Understanding its predictable periodic fluctuations could aid in developing environmental mitigation, healthcare prevention, and response preparation strategies and provide valuable insights for further epidemiological research.

Several approaches have been used to analyze $PM_{2.5}$ time series data. One method is the Prophet procedure, which decomposes time series into trend, seasonality, and holiday components and is particularly useful for forecasting with strong periodicity and flexible trend changes [7]. In addition, traditional time-series models such as ARIMA and its extensions have also been widely used to model, analyze, and forecast air pollution levels [8–10]. Many machine learning approaches have also been applied to the analysis of PM2.5 [10–12].

Bootstrapping is a resampling method thoroughly described by Efron that involves sampling with replacement from the original dataset [13]. Block bootstrapping, a variant of the bootstrapping method, resamples blocks of consecutive data points rather than individual ones. This approach is particularly useful for datasets in which observations are correlated over time. There are many different types of block bootstraps, such as the moving block bootstrap (MBB) which was proposed by Kunsch in 1989 as well as Liu and Singh in 1992, the nonoverlapping block bootstrap (NBB) which was based on the work of Carlstein in 1986 [14–16]. The block bootstrapping method, restricts the block length to be a multiple of the period $p$. Regardless of where in the cycle a block start, all other block in the resample ?> and end at a common step in the cycle with period $p$. These block bootstrap methods for periodically correlated (PC) times series are referred to here as periodic block bootstraps (PBB). The Generalized Seasonal Block Bootstrap (GSBB) is one type of PBB [17]. The PBB approach maintains the correlation structures between data points separated by $p$ intervals in the time series. In this study, we compared the Variable Bandpass Periodic Block Bootstrap (VBPBB) method with one representative PBB approach – the GSBB – to evaluate their performance in capturing periodic patterns in $PM_{2.5}$ time series.

Bootstrapping is also effective in producing confident intervals (CIs). A statistic is found from every resample in bootstrapping; therefore, the CI of this distribution can then be determined. The way of finding CIs for the block bootstrap method is similar to the general bootstrap method. The principal modification of PC bootstrapping is PBB divides the time series data into blocks of consecutive observations before repeatedly resampling these blocks with replacement.

Long-term observation of PM$_{2.5}$ is appropriate for capturing periodic characteristics, such as historical trends and seasonal changes, as well as for forecasting future tendencies. To explore the periodic characteristics of the data, PC principal components play an important role in this research. A new model-free bandpass bootstrap approach termed the VBPBB, was first introduced by Valachovic in 2024, separates PC principal components from interfering frequencies and noises by bandpass filters called Kolmogorov-Zurbenko Fourier Transforms (KZFT), and bootstraps the PC principal components [18–20]. VBPBB permits the investigation of possible periodic variation for PM2.5 and visualization of results by constructing CI bands for the periodic means of the PC components. The VBPBB method can retain the structures of the PC principal components that have been passed through the KZFT. Furthermore, this process is also capable of analyzing data with multiple periodically correlated (MPC) components, where the MPC means the presence of more than one PC component within a time series. The MPC can be seen as a function of accumulated influences or time series factors in many real-world problems. VBPBB then applies a periodic block bootstrap to the PC principal components to improve estimation of the periodic characteristics of those component processes as well as the full time series.

The purpose of this study is to explore PC principal components of daily mean levels of PM$_{2.5}$ in Manhattan and analyze this set of data by VBPBB to have a better understanding of the periodic patterns of the dataset. In a long-term observation of PM$_{2.5}$, higher PC component frequencies are readily identifiable, given the substantial number of observations in this dataset. For instance, a higher frequency might be observed every half year. Detecting these significantly higher frequency levels suggests the possibility of periodic semi-annual fluctuations in PM$_{2.5}$ levels. In this study, the results, represented by the range of the 95% CI at each time point and visualized as a 95% CI band, help distinguish between significant and insignificant higher frequencies.

## Methods

### Data sources and analysis

The dataset used in this study comprises modeled predictions of daily mean PM$_{2.5}$ concentrations in Manhattan, New York, United States, spanning from January 2001 to December 2016 seen in Fig 1. Provided by the Centers for Disease Control and Prevention (CDC), it includes 5,844 observations with no missing data [21]. This modeled prediction data was generated by Downscaler (DS) model from U.S. Environmental Protection Agency (EPA) [22]. The DS model combines monitoring data from U.S. EPA Air Quality System (AQS) repository of ambient air quality data and simulated PM$_{2.5}$ data

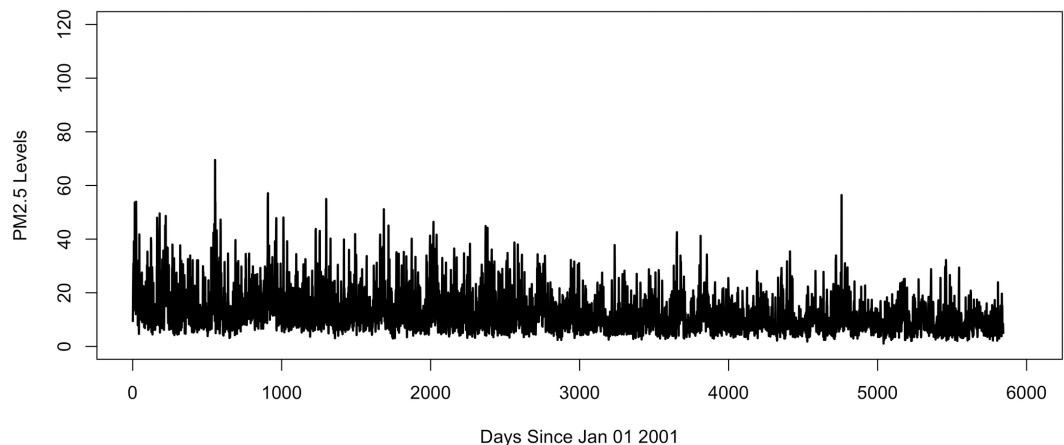

**Fig 1. Daily mean PM 2.5 levels in Manhattan from 2001 to 2016 time series.**

from Community Multiscale Air Quality (CMAQ). The CMAQ is a tool that integrates air quality science and computing to quickly estimate ozone, particulates, toxics, and other pollutants [22–24].

Our analysis is performed in R version 4.2.0 statistical software [25]. This study focuses on 6 PC components, each associated with a different frequency. Some frequencies were identified from the periodogram plot that was based on the Fourier Transform (FFT), while others were identified based on factors related to human activity and also considered their power showing on the periodogram. The periodogram is a spectral density or frequency domain representation of a time series [26]. Before analyzing the periodogram of this time series data, periodic patterns, including annual, semi-annual, and weekly, among others, were taken into consideration. As anticipated, some patterns exhibit higher powers, a phenomenon that requires validation through statistical methods, which will be demonstrated in this study. Other potential periodic characteristics will also be demonstrated in this study.

## GSBB approach

In this study, the GSBB approach is used for comparison with the VBPBB approach. The bandpass filter step is not included in the GSBB approach, which means this approach uses the original time series data comprised of noise and all other components, therefore this method resamples the unrelated components every time. In each application of the GSBB, the block length is set to the period of the PC component that is bootstrapped, such as a period of 365 of VBPBB, $p = 365$, also used to set the block length of 365 for GSBB. In each bootstrap, the number of independent resamples is set to $B = 1000$.

## Variable bandpass filter

The filter utilized in VBPBB that separates the PC principal components is a bandpass filter, known as the KZFT [20]. A bandpass filter is a process that allows PC components that have a frequency within a certain range to pass. In other words, A bandpass filter passes certain frequencies in a signal around a narrow band and suppresses frequencies outside of that band. The KZFT filter is an extension of the Kolmogorov-Zurbenko (KZ) filter which is iterated moving averages characterized by the iteration, $k$, and a positive odd integer $m$ for the filter window length. The KZ filter is denoted as $KZ_{m,k}$, and the KZFT filter is denoted as $KZFT_{m,k,v}$, where $v$ is the frequency [20,27].

When applying the KZ filter with argument $m$ and $k$ to a time series process $X(t)$ with $t = \cdots, -1, 0, 1, \cdots$, the process $KZ_{X,m,k}(t)$ is defined by following formula:

$$KZ_{X,m,k}(t) = \sum_{-\frac{k(m-1)}{2}}^{\frac{k(m-1)}{2}} a_s^{k,m} \cdot X(t+s)$$

(1)

Where $a_s^{k,m}$ are defined as: $a_s^{k,m} = \frac{C_s^{k,m}}{m^k}$, $s = \frac{-k(m-1)}{2}, \cdots, \frac{k(m-1)}{2}$

And the polynomial coefficients $C_s^{k,m}$ of $a_s^{k,m}$ are given by

$$\sum_{r=0}^{k(m-1)} z^r C_{r-k(m-1)/2}^{k,m} = \left(1 + z + \cdots + z^{m-1}\right)^k$$

The process $KZFT_{X,m,k,t}(t)$ is defined by following formula:

$$KZFT_{X,m,k,v}(t) = \sum_{-k(m-1)/2}^{k(m-1)/2} a_s^{k,m} \cdot X(t+s) \cdot e^{-i(2\pi v)s}$$

KZ filters function as symmetric low-pass filters, centered at frequency zero, effectively attenuating signals with frequencies of $1/m$ and higher, while allowing lower frequencies to pass. This process smooths the time series by preserving the trend and low-frequency components and reducing the impact of higher frequency noise. KZFT filter operates as a band-pass filter, incorporating an additional argument, $v$, making it akin to a KZ filter but centered at frequency $v$. This allows the KZFT filter to isolate a specific range of frequencies around $v$, enabling detailed analysis of periodic components within that frequency band. This work utilizes the KZFT function from the KZA package in R statistical software. Further details about the KZA package are provided in the study by Close and Zurbenko [28].

## VBPBB approach

The VBPBB approach includes the bandpass filter step. Spectral density provides a comprehensive view of where the variability of the signal is concentrated in the frequency domain, such that some potentially significant PC components of interest are found by this step. The PC components that operate at different frequencies are independent. The study evaluates 6 PC principal components corresponding to 6 distinct frequencies. These frequencies were selected based on periodogram and potential factors of human and natural activities, including the fundamental frequency at 1/365, its second harmonic and third harmonic frequencies at 2/365 and 3/365, and three additional frequencies at 1/20, 1/13, and 1/7. In addition to the frequencies of these 6 PC components of interest, the sum of significant PC components is also considered. This step entails summing the bootstrap results of the PC components.

For every PC component, the KZFT arguments are set based on the period $p_i$, and the corresponding frequencies are $v_i = 1/p_i$. The argument $k = 1$ is used in each KZFT filter which decreases data requirements. As outlined in the R package of KZA usage guide, a phase shift occurs, and the signal's fidelity decreases if the product of $v$ and $m$ is not an integer [28]. This study uses $m = 1095$ for the fundamental frequency at 1/365 and its harmonics at 2/365 and 3/365, $m = 729$ for frequencies at 1/13 and 1/7, and $m = 741$ for frequency at 1/20. Each argument $m$ was set as a multiple of the period of interest and rounded into the next odd integer.

After isolating PC components with the KZFT filter, the next step involves block bootstrapping, designed to avoid resampling unrelated components like noise or linear trends. Furthermore, the VBPBB bootstrap approach aims to resample exclusively a PC component, striving to preserve its specific correlation structure. This method effectively prevents unnecessary increases in bootstrap variability. In this study, the resample number, $B = 1000$, denotes the total number of bootstrap samples generated through sampling with replacement.

The range of 95% CI at each time point is visualized as a 95% CI band, where the upper bound is calculated by 97.5th percentiles of the bootstrap statistics, and the lower bound is calculated by 2.5th percentiles. A 95% CI band that excludes the possibility of a flat or stationary periodic mean at that frequency indicates a significant PC component at that frequency. By the process above, the critical periodic changes of $PM_{2.5}$ in Manhattan are found.

## Results

The study tests 6 PC components, which include a fundamental frequency at 1/365 and its corresponding harmonic frequencies at 2/365 and 3/365, and 3 other frequencies at 1/20, 1/13, and 1/7. Fig 2 presents the spectral density, where the x-axis represents frequency and y-axis represents power or amplitude. The results of VBPBB method show 365/2-day pattern, 365/3-day pattern, and 7-day pattern are significantly different from 0.

Fig 3 shows the bootstrapped 95% CI band for the fundamental frequency at 1/365. The GSBB is shown in red and the VBPBB is shown in blue. Across the CI band, the median GSBB CI size is 7.89 times larger than the VBPBB CI size. Both CI bands do not indicate significance, suggesting insufficient evidence to reject the hypothesis that seasonal mean variation is zero. In other words, there is not enough support for annual $PM_{2.5}$ concentration changes.

The red part is generated by GSBB approach, and the blue part is generated by VBPBB approach.

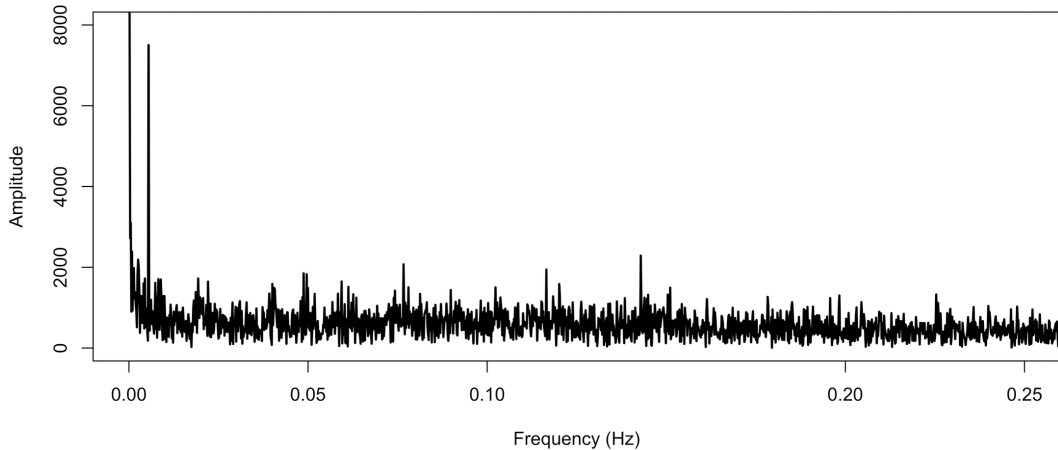

**Fig 2. The spectral density of the PM2.5 time series.**

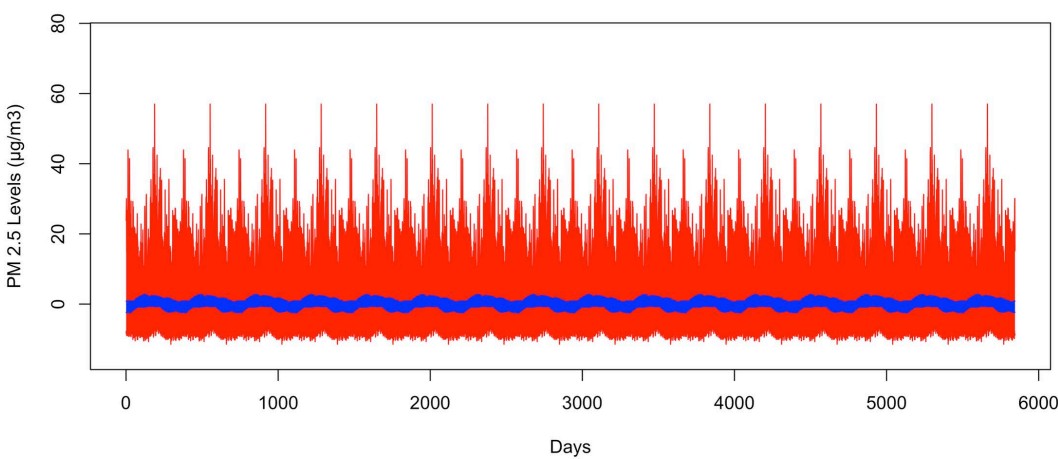

**Fig 3. The bootstrapped 95% CI bands for 365-day pattern.**

Fig 4 illustrates the bootstrapped 95% CI band for the seasonal second harmonic mean comparing GSBB in red and the VBPBB in blue across typical half-year cycles. The median GSBB CI size is 17.41 times larger than the VBPBB CI size. Fig 5 displays the VBPBB 95% CI band for the same frequency over two years, with each block representing one month. It is easy to see that there are higher levels of $PM_{2.5}$ in summer and winter and lower levels of $PM_{2.5}$ in spring and fall. The VBPBB 95% CI band provides sufficient evidence that the second harmonic is significant, rejecting that the second harmonic of the seasonal mean variation is zero.

Fig 6 illustrates the bootstrapped 95% CI band for the third harmonic mean variation across 122-day cycles (tri-annual). The GSBB and VBPBB methods are shown in red and blue, respectively. Across the CI band, the median GSBB CI size is 17.51 times larger than the VBPBB CI size. The fluctuation within the VBPBB 95% CI band for this frequency shows higher values in three periods, which include February-March, June-July, and Oct-Nov. The result of VBPBB provides sufficient evidence of significance for the third harmonic.

Fig 7 illustrates the bootstrapped 95% CI band for the mean variation over 7-day cycles. Across the CI band, the median GSBB CI size is 15.87 times larger than the VBPBB CI size. The fluctuation within the VBPBB 95% CI

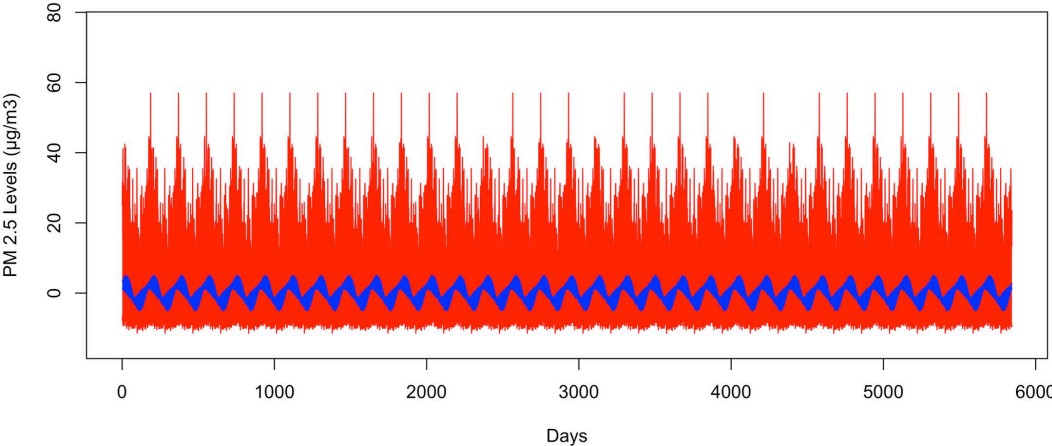

**Fig 4. The bootstrapped 95% CI band for a half-year pattern.** The red part is generated by GSBB approach, and blue part is generated by VBPBB approach.

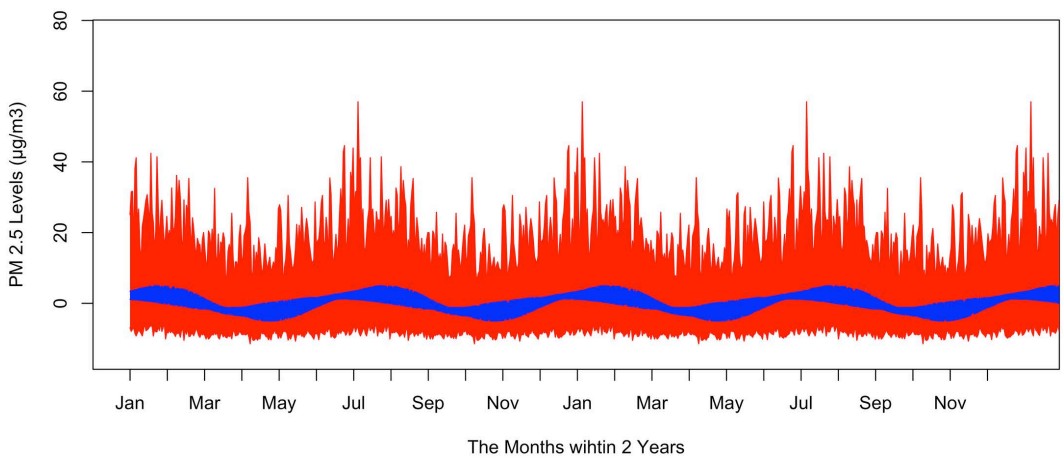

**Fig 5. The bootstrapped 95% CI band for half-year pattern for only two years.** The red part is generated by GSBB approach, and the blue part is generated by VBPBB approach. The x-axis represents daily data, and the months are labeled.

band for this frequency shows lower values on Saturdays and Sundays and higher values from Tuesday to Thursday. The VBPBB 95% CI band provides sufficient evidence that the PC principal component with frequency at 1/7 is significant.

The sum of 3 significant PC principal components is also considered. Fig 8 displays the bootstrapped 95% CI band for this combined mean variation, with the GSBB depicted in red and the VBPBB in blue. For this comparison, since GSBB cannot be applied to the sum of several periods,. The comparison uses a period of $p = 183$. The VBPBB CI band indicates that the combined component remains significant.

Table 1 summarized the bootstrapped 95% CI results VBPBB method at all tested cycles. The maximum of the 95% CIs is the range of upper bound of the band, and the minimum of the 95% CIs is the range of lower bound of the band. When 0 is included in both of these ranges, the PC component at this frequency or period is significant.

Fig 6. The bootstrapped 95% CI band for a tri-annual pattern. The red part is generated by GSBB approach, and the blue part is generated by VBPBB approach.

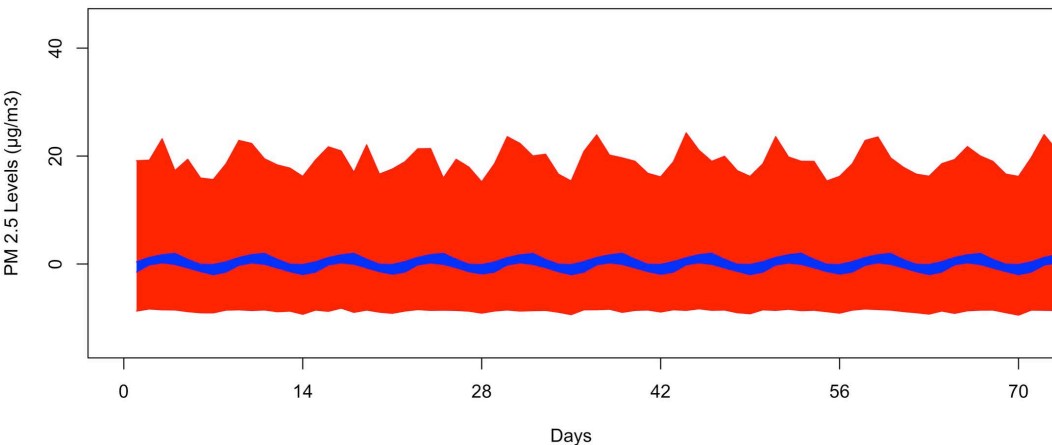

Fig 7. The bootstrapped 95% CI band for 7-day pattern. The red part is generated by GSBB approach, and the blue part is generated by VBPBB approach.

## Discussion

Long-term exposure to $PM_{2.5}$ is linked to various health issues. Understanding its periodic fluctuations could aid in developing strategies for many healthcare problems, while also offering insights for epidemiological research. Using the VBPBB, three PC principal components were identified at semi-annual, tri-annual, and weekly pattern. This suggests fluctuations in $PM_{2.5}$ concentrations during these three periods. While pinpointing the exact causes of these $PM_{2.5}$ level fluctuations falls outside this study's purview; our analysis of the 95% CI bands allows for the identification of seasonal or weekly changes in $PM_{2.5}$. In the semi-annual analysis, higher $PM_{2.5}$ concentrations were observed during the winter and summer months, whereas lower concentrations were noted in spring and fall. In the tri-annual analysis, higher $PM_{2.5}$ concentrations were observed during February-March, June-July, and October-November. In the weekly analysis, elevated $PM_{2.5}$ levels were more prevalent on weekdays, with lower levels recorded over the weekend, and pattern leads us to suspect that the weekday peaks in $PM_{2.5}$ concentrations are attributable to heavy traffic congestion in Manhattan.

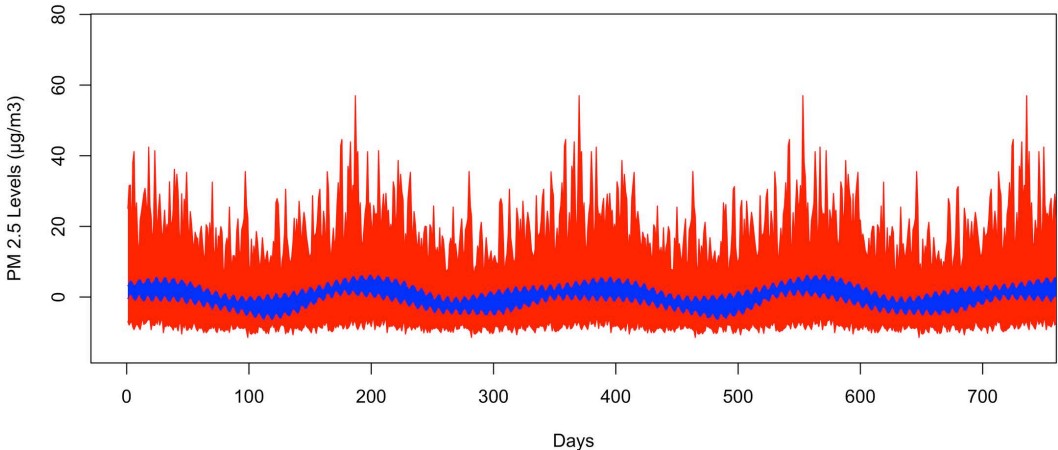

**Fig 8. The bootstrapped 95% CI band for the sum of 3 PC principal components.** The red part is generated by GSBB approach, and the blue part is generated by VBPBB approach.

**Table 1. All periods with corresponding frequencies and results tested in this study.**

| Period | Frequency | 95% CI of maximum | 95% CI of minimum |
|---|---|---|---|
| Annual | 1/365 | (0.565, 2.713) | (−2.492, −0.562) |
| Half-Annual | 2/365 | **(−1.262, 4.975)\*** | **(−4.972, 1.338)\*** |
| Tri-Annual | 3/365 | **(−0.040, 1.918)\*** | **(−1.765, 0.022)\*** |
| Every 20 days | 1/20 | (0.238, 2.101) | (−2.210, −0.303) |
| Every 13 days | 1/13 | (0.404, 2.066) | (−2.039, −0.334) |
| Every 7 days | 1/7 | **(−0.116, 2.007)\*** | **(−2.011, 0.237)\*** |

\* P < 0.05

Our finds are consistent with the results reported by Zhao et al., who used EPA ground-based monitoring data [7]. Specifically, their summary of $PM_{2.5}$ trends in New York City shows higher concentrations during winter and summer, which aligns well with the seasonal patterns detected by our method. Notably, in their findings, unlike many other U.S. cities where weekends peaks are observed, New York City exhibits a weekday peak in $PM_{2.5}$, and our analysis captured this same weekly pattern as well.

There are advantages of VBPBB. One advantage of the VBPBB is its capability to yield significant results. The VBPBB is capable of generating both the 95% CIs and the 95% CI bands following the periodic block bootstrapping steps. This enables the identification and visualization of fluctuations in any PC principal components of interest through this process. Once the 95% CIs are produced, the significance can be detected. This step is an advancement over the GSBB, as the 95% CI band in the latter contains excessive noise and unwanted frequencies, making it challenging to determine significance.

Another advantage of VBPBB is its capability to sum up all PC components of interest. As demonstrated in Fig 8, when the three significant PC components are combined, the 95% CI band is correspondingly updated. This feature allows for a comprehensive analysis of the factors that affect levels of $PM_{2.5}$. This combined 95% CI band can also be useful for forecasting the future fluctuation or periodic characteristics of $PM_{2.5}$.

There are also limitations in this study. One limitation of the VBPBB is that the method used KZFT filter to pass certain frequencies, and different arguments defined by KZFT filter are used. The effectiveness of this approach hinges on

the selection of different arguments ($k$, $m$, and $v$) defined by the KZFT filter, with the study's results varying according to the chosen arguments for each frequency. In most real-world applications of the KZFT filter, the argument for iterations, $k$, is typically set to 1 or 2 because larger values of $k$ result in larger data requirements, potentially leading to a loss of critical details that could affect the outcomes. Consequently, this study opts for $k = 1$ across all frequencies, deeming $k = 2$ unnecessary for this dataset given that the fluctuations in PM$_{2.5}$ levels in Manhattan from 2001 to 2016 are not pronounced. Another crucial argument is the window width, $m$, and the larger the value the narrower the bandpass filter resulting in more uncorrelated frequencies excluded. To illustrate, if $m = 1$ and $k = 1$ are chosen for the KZFT filter, the VBPBB reduces to that of the GSBB, capturing many details but failing to ascertain their significance. The VBPBB will benefit from additional research into the optimal choice of KZFT arguments, and results obtained here may further improve with a different selection of arguments.

The study utilized daily mean levels of PM$_{2.5}$ in Manhattan as an example to apply the VBPBB method. However, this approach is not confined to Manhattan nor exclusively to PM$_{2.5}$ data. It demonstrates a versatile framework that can be adapted for various geographical locations and environmental pollutants or other fields. This study focuses on Manhattan as an initial application of the VBPBB method, given its dense population, heavy traffic, and distinct seasonal variation. In the future work, the analysis may be expanded to include additional urban areas with varying climatological or emission characteristics, as well as to comparisons between urban and rural regions, aiming to improve understanding of spatial difference in PM$_{2.5}$ temporal patterns. Future studies may also consider applying the same methodology to PM$_{10}$ time series to examine whether similar periodic patterns emerge. This adaptability underscores the potential of the VBPBB method to analyze a wide range of time series data, offering insights into environmental trends and health implications.

The method employed in this study is highly appropriate for the dataset. This study encompasses 5,844 observations over 16 years. This abundance of data proves especially valuable for investigating PM$_{2.5}$ concentration fluctuations on an annual, seasonal, and weekly basis, as well as during other specific periods. These results would benefit with a longer record of PM$_{2.5}$ to investigate longer period components, like changes over multiple years or decades. Also, finer records would permit investigation of PM$_{2.5}$ periodicity on scales less than weekly. In this study, three significant PC components of PM$_{2.5}$ were identified, along with one significant sum of PC components. These three independent PC components are crucial for understanding the dynamics of PM$_{2.5}$ concentrations. Their identification enhances the comprehension of the factors influencing air quality, provides insights for other studies on air pollutants, and informs future healthcare research related to diseases associated with PM$_{2.5}$.

## Acknowledgments

A preprint has previously been published [29].

## Author contributions

**Data curation:** Yanan Sun.

**Formal analysis:** Yanan Sun.

**Investigation:** Yanan Sun.

**Methodology:** Yanan Sun.

**Software:** Yanan Sun.

**Supervision:** Edward L Valachovic.

**Visualization:** Yanan Sun.

**Writing – original draft:** Yanan Sun.

**Writing – review & editing:** Edward L Valachovic.

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
