## [Decision Letter · Decision Letter 0]

PONE-D-25-05546Seasonal and Periodic Patterns of PM2.5 in Manhattan using the Variable Bandpass Periodic Block BootstrapPLOS ONE

Dear Dr. Sun,

Thank you for submitting your manuscript to PLOS ONE. After careful consideration, we feel that it has merit but does not fully meet PLOS ONE’s publication criteria as it currently stands. Therefore, we invite you to submit a revised version of the manuscript that addresses the points raised during the review process.

We look forward to receiving your revised manuscript.

Kind regards,

Loredana Bellantuono, Ph.D.

Academic Editor

PLOS ONE

Reviewers' comments:

Reviewer's Responses to Questions

**Comments to the Author**

1. Is the manuscript technically sound, and do the data support the conclusions?

Reviewer #1: Yes

Reviewer #2: Yes

2. Has the statistical analysis been performed appropriately and rigorously? 

Reviewer #1: Yes

Reviewer #2: Yes

3. Have the authors made all data underlying the findings in their manuscript fully available?

Reviewer #1: Yes

Reviewer #2: Yes

4. Is the manuscript presented in an intelligible fashion and written in standard English?

Reviewer #1: Yes

Reviewer #2: Yes

5. Review Comments to the Author

Reviewer #1: In this study, Sun et al. propose an innovative bandpass bootstrap approach to extract temporal patterns from PM2.5 time series. While the methodological framework is original and the manuscript is generally well written with a clear structure, several aspects would benefit from further clarification or extension:

1) The authors focused on the temporal behavior of PM2.5 from 2001 to 2016. Have the authors considered applying the same approach to PM10 time series, if available? A comparative analysis could reveal whether similar or divergent temporal patterns emerge between PM2.5 and PM10, thereby strengthening the robustness and interpretability of the findings.

2) The PM2.5 concentrations used in the analysis are derived from the CMAQ model, which might introduce biases into the extracted patterns. Have the authors attempted to perform a similar analysis using only ground-based monitoring data, possibly by filtering out missing values or limiting the analysis to a shorter but fully observed time window? This could help assess the influence of modeled data on the detected patterns.

3) The study focuses exclusively on the Manhattan area. This spatial limitation may affect the generalizability of the results. While the authors suggest that their approach can be extended to other regions, it would strengthen the manuscript to include a validation case in a different urban area, preferably one with distinct climatological or emission characteristics, and compare the extracted patterns.

4) Once the temporal patterns have been extracted, it would be important to contextualize them with existing literature. Are the results consistent with previously reported seasonal or long-term trends? A discussion in this direction would help validate the method and highlight novel contributions.

Minor comments:

1) The use of artificial intelligence methods could be a promising direction to extend this work, especially when dealing with larger datasets or diverse geographic regions. Have the authors explored the potential of approaches such as Recurrent Neural Networks (RNNs), Long Short-Term Memory networks (LSTMs), or Transformers for pattern extraction? These models may offer valuable insights or complementary perspectives.

2) The language in the Results section is somewhat repetitive in certain parts. The readability and flow of the manuscript could benefit from stylistic variation.

Overall, the manuscript presents a promising and novel approach to temporal pattern analysis in air pollution data. With the suggested improvements, it could make a strong contribution to the field.

Reviewer #2: I believe the objective pursued by this work is well-aligned with the research questions and the current open issues in this field. Increasingly, researchers are investigating the identification of temporal patterns—seasonal, annual, and weekly—in PM2.5 concentrations, aiming to trace back the underlying causes and pollution sources. These may include traffic emissions, pollen, anthropogenic forcing, or natural sources such as desert dust. The approach adopted is reasonable, although it should be noted that during accidental or exceptional events, recorded measurements may deviate from typical periodic patterns.

1. Introduction and Literature Review.The introduction initially focuses heavily on the epidemiological aspect, while it appears to lack sufficient bibliographic references related to the application of various bootstrapping techniques to time series of PM or analogous particulate measurements. Although this methodological component is outside my domain of expertise, the techniques used to assess the stability of the frequency spectrum—highlighting key periodicities with associated confidence intervals—appear to be mathematically well-formulated. I will refrain from commenting further on this part.

However, I recommend including more literature in both the introduction and discussion sections, particularly regarding applied case studies involving PM2.5 time series analysis and their associated findings.

2. Clarification on CMAQ Application Scope. Is the Community Multi-Scale Air Quality (CMAQ) model only applicable within the United States? If so, this limitation should be clearly specified. Otherwise, readers might incorrectly assume that it can provide high-resolution pollutant concentration estimates “globally” for any emission or climate scenario.

3. Use of Ground-Based Monitoring Data. Would it not have been appropriate to include, for comparison purposes, data acquired from air quality monitoring stations located within the Manhattan area? Or were the data used in this study already sourced from such stations?

In either case, a brief mention or justification regarding the inclusion or exclusion of these ground-based monitoring data would be appreciated.

4. Figure Readability. The placement of the figures at the end of the manuscript hinders a smooth reading flow and makes it more difficult to follow the interpretation of the results. Integrating the figures more closely with the corresponding text would significantly enhance readability and comprehension.

6. PLOS authors have the option to publish the peer review history of their article (what does this mean? ). If published, this will include your full peer review and any attached files.

**Do you want your identity to be public for this peer review?** For information about this choice, including consent withdrawal, please see our Privacy Policy .

Reviewer #1: **Yes: ** Alessandro Fania

Reviewer #2: No

---

## [Author Response · Author response to Decision Letter 1]

22 May 2025

Dear Editor and Reviewers,

Please allow me to thank you all for your time, reviews, comments, and suggestions. It is my hope and endeavor that my responses and changes address each comment to our mutual satisfaction. I will address each point within a copy of the email I received (below), organized in the sections specific to each of you, with corresponding responses (in bold) that reference the marked-up manuscript and tracked changes (see file ‘'Revised Manuscript with Track Changes’). Thank you again.

Sincerely, Yanan Sun

PONE-D-25-05546

Seasonal and Periodic Patterns of PM2.5 in Manhattan using the Variable Bandpass Periodic Block Bootstrap

PLOS ONE

Dear Dr. Sun,

Thank you for submitting your manuscript to PLOS ONE. After careful consideration, we feel that it has merit but does not fully meet PLOS ONE’s publication criteria as it currently stands. Therefore, we invite you to submit a revised version of the manuscript that addresses the points raised during the review process.

We look forward to receiving your revised manuscript.

Kind regards,

Loredana Bellantuono, Ph.D.

Academic Editor

PLOS ONE

Thanks for your feedback. I have reviewed the formatting guidelines provided by PLOS ONE and made some changes.

The data are publicly available, are cited in the manuscript, and can be accessed at: https://data.cdc.gov/Environmental-Health-Toxicology/Daily-County-Level-PM2-5-Concentrations-2001-2019/dqwm-pbi7/about_data

I uploaded to public repository: DOI: https://doi.org/10.6084/m9.figshare.29132738

Reviewers' comments:

Reviewer's Responses to Questions

Comments to the Author

1. Is the manuscript technically sound, and do the data support the conclusions?

Reviewer #1: Yes

Reviewer #2: Yes

2. Has the statistical analysis been performed appropriately and rigorously?

Reviewer #1: Yes

Reviewer #2: Yes

3. Have the authors made all data underlying the findings in their manuscript fully available?

Reviewer #1: Yes

Reviewer #2: Yes

4. Is the manuscript presented in an intelligible fashion and written in standard English?

Reviewer #1: Yes

Reviewer #2: Yes

5. Review Comments to the Author

Reviewer #1: In this study, Sun et al. propose an innovative bandpass bootstrap approach to extract temporal patterns from PM2.5 time series. While the methodological framework is original and the manuscript is generally well written with a clear structure, several aspects would benefit from further clarification or extension:

1) The authors focused on the temporal behavior of PM2.5 from 2001 to 2016. Have the authors considered applying the same approach to PM10 time series, if available? A comparative analysis could reveal whether similar or divergent temporal patterns emerge between PM2.5 and PM10, thereby strengthening the robustness and interpretability of the findings.

Thank you for your insightful suggestion regarding the potential application of our approach to PM10 time series. I agree that a comparative analysis between PM2.5 and PM10 could provide valuable insights into whether similar or divergent temporal patterns exist, thereby enhancing the robustness and interpretability of our findings. I added a statement in the Discussion section of revised manuscript to describe this and recommend for a future research project.

2) The PM2.5 concentrations used in the analysis are derived from the CMAQ model, which might introduce biases into the extracted patterns. Have the authors attempted to perform a similar analysis using only ground-based monitoring data, possibly by filtering out missing values or limiting the analysis to a shorter but fully observed time window? This could help assess the influence of modeled data on the detected patterns.

Thank you for your suggestion. This dataset is the combination of PM2.5 monitoring data from the US EPA Air Quality System (AQS) repository of ambient air quality and model predicted PM2.5 data from the CMAQ, and Downscaler (DS) model provided the output for this combination. To improve clarity, I revised the relevant parts of the manuscript.

Based on the website of US CDC, these data are used by the CDC’s National Environmental Public Health Tracking Network to generate air quality measures. I agree with you that data used in the analysis is derived from the CMAQ model, which might introduce biases into the extracted patterns, but the dataset we used combined the monitoring data. As a result, the risk of bias is reduced, and no missing value is included, and we believe this is a good data source to evaluate the VBPBB method. Furthermore, the VBPBB has been proven to be robust against biases based on our other studies.

We also reviewed other studies that include PM2.5 data for New York State and New York City, including one that used ground-based monitoring data. Preliminary results were generally consistent with ours, suggesting that although CMAQ-modeled data may introduce some bias, the impact on the detected temporal patterns appears to be limited. This referenced study, which used ground-based monitoring data in New York City, has been cited and discussed in the revised Discussion section to further support the consistency and reliability of our study. In the future, we may also consider using the VBPBB approach to compare DS PM2.5 data with ground-based observations.

3) The study focuses exclusively on the Manhattan area. This spatial limitation may affect the generalizability of the results. While the authors suggest that their approach can be extended to other regions, it would strengthen the manuscript to include a validation case in a different urban area, preferably one with distinct climatological or emission characteristics, and compare the extracted patterns.

I appreciate this thoughtful suggestion. We agree that including a comparison with another urban area featuring distinct climatological or emission characteristics would enhance the manuscript. We initially selected Manhattan because it is a well-known urban area with high population and traffic density, as well as clear seasonal variation. This study serves as a strong starting point for applying the VBPBB method in environmental time series analysis. As part of our future work, we plan to expand the analysis to include comparisons across different cities, and also between urban and rural areas. A spatial VBPBB application is in development. These comparisons will help us gain a more comprehensive understanding of PM2.5 dynamics across diverse settings. I have added a statement in the Discussion section to recommend this.

4) Once the temporal patterns have been extracted, it would be important to contextualize them with existing literature. Are the results consistent with previously reported seasonal or long-term trends? A discussion in this direction would help validate the method and highlight novel contributions.

Thanks for the suggestion. I added new paragraph in the Discussion section for this.

Minor comments:

1) The use of artificial intelligence methods could be a promising direction to extend this work, especially when dealing with larger datasets or diverse geographic regions. Have the authors explored the potential of approaches such as Recurrent Neural Networks (RNNs), Long Short-Term Memory networks (LSTMs), or Transformers for pattern extraction? These models may offer valuable insights or complementary perspectives.

Thank you for the thoughtful suggestion. I have explored recurrent models such as RNNs and LSTMs, as well as Transformer-based architectures. Recently, I have been placing more focus on Transformer models. However, I have not yet identified a meaningful way to integrate these models with the VBPBB framework in a manner that complements its frequency-domain strengths.

That said, I agree that attention-based models like Transformers could offer valuable insights – particular through the use of attention maps to highlight which time points the model considered most influential. This could provide an alternative and highly interpretable perspective on temporal patterns in PM2.5 data. I recognized that Transformer models and the VBPBB method emphasize different aspects of time series analysis. One focuses on data-driven temporal attention, while the other highlights periodic structure through frequency-domain decomposition. Although I have not yet identified a direct way to integrate them meaningfully, I believe this is a promising area for future exploration. A thoughtful combination of the two could help extract richer patterns, analyze complicated data, and provide insights that are beneficial for broader scientific research.

Additionally, if such a data or study become available in the future, particularly with a larger set of features, we may consider applying the approaches mentioned above.

2) The language in the Results section is somewhat repetitive in certain parts. The readability and flow of the manuscript could benefit from stylistic variation.

Thanks for the suggestion. I edited some in the revised version.

Overall, the manuscript presents a promising and novel approach to temporal pattern analysis in air pollution data. With the suggested improvements, it could make a strong contribution to the field.

Reviewer #2: I believe the objective pursued by this work is well-aligned with the research questions and the current open issues in this field. Increasingly, researchers are investigating the identification of temporal patterns—seasonal, annual, and weekly—in PM2.5 concentrations, aiming to trace back the underlying causes and pollution sources. These may include traffic emissions, pollen, anthropogenic forcing, or natural sources such as desert dust. The approach adopted is reasonable, although it should be noted that during accidental or exceptional events, recorded measurements may deviate from typical periodic patterns.

1. Introduction and Literature Review.The introduction initially focuses heavily on the epidemiological aspect, while it appears to lack sufficient bibliographic references related to the application of various bootstrapping techniques to time series of PM or analogous particulate measurements. Although this methodological component is outside my domain of expertise, the techniques used to assess the stability of the frequency spectrum—highlighting key periodicities with associated confidence intervals—appear to be mathematically well-formulated. I will refrain from commenting further on this part.

However, I recommend including more literature in both the introduction and discussion sections, particularly regarding applied case studies involving PM2.5 time series analysis and their associated findings.

Thanks for the suggestion. I modified the introduction part and added additional citations and references. I agree with you that it is better to add some details about PM2.5 time series analysis and their associated findings.

2. Clarification on CMAQ Application Scope. Is the Community Multi-Scale Air Quality (CMAQ) model only applicable within the United States? If so, this limitation should be clearly specified. Otherwise, readers might incorrectly assume that it can provide high-resolution pollutant concentration estimates “globally” for any emission or climate scenario.

Thanks for pointing this out. The PM2.5 data used in this study were obtained from U.S. CDC, and as such, the dataset includes only U.S. cities. However, we would like to clarify that the CMAQ model itself is not limited to the United States. For example, the paper titled “One-year simulation of ozone and particular matter in China using WRF/CMAQ modeling system” demonstrat

---

## [Decision Letter · Decision Letter 1]

Seasonal and Periodic Patterns of PM2.5 in Manhattan using the Variable Bandpass Periodic Block Bootstrap

PONE-D-25-05546R1

Dear Dr. Sun,

We’re pleased to inform you that your manuscript has been judged scientifically suitable for publication and will be formally accepted for publication once it meets all outstanding technical requirements.

Kind regards,

Loredana Bellantuono, Ph.D.

Academic Editor

PLOS ONE

---

## [Editor Report · Acceptance letter]

PONE-D-25-05546R1

PLOS ONE

Dear Dr. Sun,

I'm pleased to inform you that your manuscript has been deemed suitable for publication in PLOS ONE. Congratulations! Your manuscript is now being handed over to our production team.

Kind regards,

on behalf of

Prof. Loredana Bellantuono

Academic Editor

PLOS ONE